# Transcriptome analysis in rhesus macaques infected with hepatitis E virus genotype 1/3 infections and genotype 1 re-infection

**Youkyung H. Choi** [1]*, **Xiugen Zhang**[1], **Ganesh Srinivasamoorthy**[2], **Michael A. Purdy**[1]

**1** Division of Viral Hepatitis, National Center for HIV/AIDS, Viral Hepatitis, STD and TB Prevention, The Centers for Disease Control and Prevention, Atlanta, Georgia, United States of America, **2** Office of Advanced Molecular Detection, National Center for Emerging and Zoonotic Infectious Diseases, The Centers for Disease Control and Prevention, Atlanta, Georgia, United States of America

\* brt5@cdc.gov

**Data Availability Statement:** All of the transcriptome data obtained in this study have been submitted to NCBI the Sequence Read Archive (SRA) under accessionPRJNA648087and submission ID SUB7770016.

## Abstract

Hepatitis E virus (HEV) genotype 1 (gt1) and gt3 infections have distinct epidemiologic characteristics and genotype-specific molecular mechanisms of pathogenesis are not well characterized. Previously, we showed differences in immune response-related gene expression profiles of HEV gt1 and gt3 infections using qPCR. We hypothesize that HEV gt1 and gt3 infections induce transcriptome modifications contributing to disease pathogenesis. RNA-seq analysis was performed using liver biopsy samples of naïve (baseline), HEV gt1, or gt3-infected rhesus macaques, and nine anti-HEV positive rhesus macaques re-inoculated with HEV gt1. All 10 primary HEV gt1/gt3 infected animals exhibited the typical course of acute viral hepatitis and cleared the infection between 27 to 67 days after inoculation. Viremic stages of HEV infection were defined as early, peak, and decline based on HEV RNA titers in daily stool specimens. During early, peak, and decline phases of infection, HEV gt1 induced 415, 417, and 1769 differentially expressed genes, respectively, and 310, 678, and 388 genes were differentially expressed by HEV gt3, respectively (fold change $\geq$ 2.0, $p$-$value \leq 0.05$). In the HEV gt1 infection, genes related to metabolic pathways were differentially expressed during the three phases of infection. In contrast, oxidative reduction (early phase), immune responses (peak phase), and T cell cytokine production (decline phase) were found to be regulated during HEV gt3 infection. In addition, FoxO and MAPK signaling pathways were differentially regulated in re-infected and protected animals against HEV gt1 reinfection, respectively. Significant differences of hepatic gene regulation exist between HEV gt1 and gt3 infections. These findings reveal a new link between molecular pathogenesis and epidemiological characteristics seen in HEV gt1 and gt3 infections.

## Introduction

Hepatitis E, a liver disease caused by hepatitis E virus (HEV) is a public health problem and a major cause of enterically transmitted acute viral hepatitis worldwide [1]. In 2012, HEV was

**Funding:** Funding source of this study was financial support by National Center for HIV/AIDS, Viral Hepatitis, STD, and TB Prevention (NCHHSTP), CDC. The funders had no role in study design, data collection and analysis, decision to publish, or preparation of the manuscript.

**Competing interests:** The authors have declared that no competing interests exist.

estimated to cause 20 million infections worldwide including 3.3 million symptomatic cases and 56,600 HEV-related deaths [2, 3].

HEV causes both epidemic and sporadic viral hepatitis [4–6] HEV (*Hepeviridae Orthohepevirus A*) is classified into eight genotypes (gt): gt1 and gt2 only infect humans; gt3 and gt4 circulate in animals including swine, boars, rabbits, deer, and mongooses as well as humans; gt5 and gt6 were identified in Japanese wild boars; gt7 and gt8 were found in dromedary and Bactrian camels, respectively [7–9]. HEV infections in humans (gt1-4) have distinctive geographic distribution and contrast epidemiologically. HEV gt1 and gt2 infections are prevalent in parts of Asia, Africa, and the Middle East where they are transmitted through the fecal-oral route, while gt3 and gt4 infections are found mainly in industrialized countries and cause sporadic zoonotic foodborne hepatitis E [2, 10]. HEV gt1 and gt2 infections are seen primarily in adolescents and young adults and are more common in men than women. These infections can lead to fulminant liver failure, particularly in pregnant women [11–13]. The mortality rate in pregnancy is 20% to 25%, and maternal death usually occurs in the third trimester [13]. The seroprevalence rate of anti-HEV antibodies in most parts of Asia and Africa is 10%–40% [14]. HEV gt3 and gt4 infections are mostly asymptomatic, with symptomatic hepatitis found primarily in middle-aged and elderly men [6, 15, 16]. Although HEV causes a self-limited disease, gt3 infection can lead to chronic hepatitis E in immunosuppressed patients following organ transplantation, individuals co-infected with HIV, and hematological patients [17–19]. In addition, extra-hepatic manifestations associated with ongoing or previous HEV infections are reported [20].

Molecular mechanisms of pathogenesis related to genotype-specific HEV infection are not clear. In previous studies, we reported on hepatic expression profiles of immune response-related gene expression against primary HEV gt1 and gt3 infections and homologous HEV gt1 re-infection in experimentally infected rhesus macaques using qPCR with a limited set of genes [21, 22]. In this study we applied next-generation sequencing to liver biopsies collected from rhesus macaques, with primary HEV gt1 and 3 infections, and animals re-inoculated with gt1, to analyze the transcriptomes of these macaques. Gene ontology (GO) terms and pathway analysis were performed to gain insight into molecular mechanisms against genotype-specific primary and secondary HEV infections.

## Materials and methods

### Ethical statement and clinical care of animals

Procedures performed for research protocols (2364CHOMONC, 2693CHOMONC, 2643CHOMONC) followed all federal regulatory requirements for animal care and use and were approved by the Institutional Animal Care and Use Committee at the Centers for Disease Control and Prevention (CDC) in Atlanta, GA. No animals were euthanized for this study. Disposition of animals at the end of study was transfer to another principal investigator at CDC. Routine clinical care consisted of a comprehensive annual examination, a semiannual examination and a twice daily observation for clinical and psychological abnormalities. The annual comprehensive examination included a physical examination hematology, clinical chemistry, bacteriology analysis (screening for *Mycobacterium tuberculosis*, *Shigella*, *Salmonella*, *Campylobacter*, and *Yersinia*), and virology antibody titers (Hepatitis E, measles, *Macacine herpesvirus 1*, simian T-cell leukemia virus type 1, and simian retrovirus type D, simian immunodeficiency virus) and simian retrovirus by PCR. The semiannual examination consisted of a physical examination, tuberculosis palpebral test, hematology, and clinical chemistry. Animal were monitored twice daily for any physical illness or psychological distress with

abnormalities reported to the appropriate veterinary clinical staff and research staff according to facility standard operating procedures.

## Animal housing and enrichment

Animal housing at CDC was in accordance with *The Guide for the Care and Use of Laboratory Animals* in a facility accredited by the Association for Assessment and Accreditation of Laboratory Animal Care International. All animals were singly housed indoors in one-over-one stainless steel cages with a floor area space of 6.0 ft$^2$ (0.56 m$^2$) and a height of 32 in. (81.3 cm). Single housing occurred for eight days before inoculation (collection of control specimens) and after inoculation to monitor HEV shedding in each individual animal. Animals remained singly housed until viral shedding ceased at time the animals could be socially housed. Environmental conditions were maintained at a relative humidity and temperature of 30%– 70% and 64–84 ˚F (18–29 ˚C), respectively, under a 12:12-h light: dark cycle. Water was provided ad libitum through an automatic delivery system. The diet comprised of a standard monkey chow (Lab Diet Monkey Diet 5038, PMI$^®$ Nutrition International), fruits and vegetables, and commercial treats (Bio-serv, Flemington, NJ). Environmental enrichment consisted of various toys and manipulanda items, cage complexities, foraging and task-oriented feeding methods, and human interaction.

## Laparoscopic liver biopsy procedures

Animals were sedated with 10 mg/kg ketamine hydrochloride (Ketalar$^®$ Parke-Davis, Detroit, Michigan) injected intramuscularly. Anesthesia was induced by isoflurane (Isothesia, Henry Schein, Melville, NY) at 5% initially and then maintained by using a gas flow rate of 1% to 3%. Vital signs were monitored with monitoring equipment (Dre Medical International, Louisville, KY). A liver biopsy was performed aseptically once on two animals prior to hepatitis E virus inoculation to observe baseline liver tissue for host response induced by hepatitis E virus infection using a one-port incisional abdominal entry method with a laparoscope (Biovision Veterinary Endoscopy, LLC., Denver, CO). The liver was visualized with the laparoscope, and a biopsy sample was obtained from a liver lobe with the laparoscopic biopsy instrument. Hemorrhage was assessed and controlled appropriately with applied pressure from biopsy instrument, bipolar cautery, or with sterile absorbable compressed sponge (Gelfoam$^®$ Pfizer, New York, New York). Abdominal incision for port entry was closed with 3–0 polyglactin 910 (Ethicon, Somerville, NJ) and skin apposition with surgical glue (SutureVet VetClose, Henry Schein, Melville, NY) reinforced with simple interrupted sutures of 3–0 nylon (Ethicon, Somerville, NJ).

## Percutaneous needle liver biopsy procedure

Animals were sedated with either tiletamine and zolazepam 3–5 mg/kg intramuscularly (Telazol$^®$, Zoetis Inc., Kalamazoo, MI) or ketamine/xylazine 10mg/kg and 0.5mg/kg intramuscularly. An ultrasound-guided needle liver biopsy specimen was obtained from rhesus monkeys infected with HEV genotype 3 to explore host gene regulation induced by HEV genotype 3 infection at two different times during the study. Using the aseptic technique, a percutaneous needle liver biopsy was performed after HEV genotype 3 inoculation. Only three liver biopsies were allowed for collection of liver tissue and was performed by only trained personnel. A 16 or 18-gauge disposable core biopsy Instrument (BARD$^®$ MAX-CORE$^®$ Tempe, AZ) was inserted through the skin, into the abdomen, and guided up to the liver with the visualization of an ultrasound probe. The needle was retracted and advanced to cut the desired liver tissue sample. The needle was then withdrawn from the liver and abdomen and the specimen was

obtained for diagnostics. The first liver biopsy was performed at peak of HEV RNA titer detected in stool, and the second liver biopsy was during the final week of the study (105 or 115 days after inoculation). If the quality of total RNA from the liver biopsied tissue is inadequate for gene expression profiling applications, a second liver biopsy was performed a week after the first attempt of the liver biopsy. For analgesia, the animals receive 0.5–1.0 ml of lidocaine subcutaneously in area of biopsy site immediately after the biopsy was performed, 0.01mg/kg of buprenorphine intramuscularly for 3 days and 0.1–0.3mg/kg of meloxicam orally or intramuscularly for three days. Animals were assessed for clinical signs associated with loss of appetite, pain, or any signs of abdominal complications associated with the liver biopsy procedure.

### Inoculum and blood sampling

For blood collection and virus inoculation, animals were anesthetized with 10 mg/kg ketamine hydrochloride (Ketalar® Parke-Davis, Detroit, Michigan) injected intramuscularly. Blood samples were collected from either the inguinal or saphenous veins with a 1 inch 21-23-gauge needle. Three rhesus macaques (RH623, RH620, and RH625) were inoculated intravenously with a pooled stool suspension from rhesus macaques infected with the human HEV gt1 Sar-55 strain [21]. Seven animals (RH654, RH641, RH644, RH639, RH645, RH642, and RH650) were inoculated intravenously with a pooled stool suspension from rhesus macaques infected with a human HEV gt3 strain (GenBank access number, JN837481) [23]. Nine animals that had previously cleared a primary HEV gt1 Sar-55 infection [21] were re-inoculated intravenously with homologous HEV gt1 Sar-55 (Table 1). All the animals were monitored for 115 days after inoculation.

### Determination of HEV RNA titer by real-time PCR

Total RNA extraction was either performed using the RiboPure RNA purification Kit (Life Technology, Carlsbad, CA) or the Total Nucleic Acid Isolation Kit (Roche Applied Science, Indianapolis, IN) on the MagNA Pure LC 2.0 instrument (Roche Applied Science) according

**Table 1. Liver biopsy sample of the animals with primary HEV gt1/gt3 infection used for RNA sequencing.**

| animal ID | HEV | Viremic stage | HEV RNA (Log$_{10}$ WHO IU/ml*) | | | ALT activity/ cutoff (U/L) | Anti-HEV AB | |
|---|---|---|---|---|---|---|---|---|
| | | | liver | stool | serum | | IgM | IgG |
| RH623 | gt1 | early | 8.8 | 7.2 | 5.3 | 34/54 | NEG | NEG |
| RH620 | gt1 | peak | 9.7 | 9.1 | 5.1 | 46/53 | POS | POS |
| RH625 | gt1 | decline | 4 | 4 | NEG | 113/40 | POS | POS |
| RH654 | gt3 | early | 6.3 | 4 | NEG | 35/44 | NEG | NEG |
| RH641 | gt3 | early | 5.2 | 5 | 1.9 | 35/43 | NEG | NEG |
| RH644 | gt3 | peak | 8.8 | 8.9 | 3.7 | 231/71 | NEG | NEG |
| RH639 | gt3 | peak | 7.7 | 9.3 | 3.5 | 65/36 | POS | NEG |
| RH645 | gt3 | peak | 7 | 7.9 | 4 | 110/36 | POS | POS |
| RH642 | gt3 | peak | 3.7 | 5.8 | 1.8 | 38/29 | POS | POS |
| RH650 | gt3 | decline | 4.1 | 7.3 | 1.7 | 308/58 | POS | POS |

*, detection limit of HEV RNA by real-time PCR is 20 WHO IU/ml. The course of infection has been described previously (Choi et al., 2018). Viremic stages is defined based on HEV RNA titer in stool samples. GT1, genotype 1; GT3, genotype 3; NEG, negative detection of anti-HEV antibody responses; POS, positive detection of anti-HEV antibody responses.

A pooled stool suspension from rhesus macaques with HEV gt1 or gt3 infection, ranging from 2.5 to 9.5 Log$_{10}$ WHO IU/ml was used for gt1 or gt3 infection (Choi et al, 2018).

to the manufacturer's recommended procedure. The HEV RNA titer in liver, stool and serum specimens was determined as an RNA value in IU/mL by comparison with a standard curve of serial $\log_{10}$ dilutions of HEV with known potency in WHO international units [24].

### Detection of anti-HEV antibodies and ALT activity

Levels of alanine aminotransferase activity (ALT) and anti-HEV antibodies were measured as described earlier [21, 22]. Multiple pre-inoculation, weekly post-inoculation sera collected from the animals were tested for ALT activity using a VetScan VS2 (ABAXIS, Union City, CA) and anti-HEV antibody response using commercially available enzyme immunoassays (DSI, Milan, Italy) performed in accordance with manufacturer's instructions, and recorded as signal-to-cutoff ratios.

### Sample preparation

Total RNA was extracted from liver biopsy specimens of two naïve control (pre HEV infection) and HEV gt1- or gt3-infected animals (post HEV infection) using the Invitrogen Ribo-Pure kit (ThermoFisher, Waltham, MA). mRNA was isolated with NEBNext Poly(A) mRNA Magnetic Isolation kit (New England BioLabs, Ipswich, MA). NEBNext Ultra II RNA Library Prep kit for Illumina (New England BioLabs) was used to generate RNAseq libraries according to manufacturer's protocol. The quality of the RNAseq libraries was analyzed on a Bioanalyzer 2100 (Agilent, Santa Clara, CA). Libraries were sequenced as 100 to 200-bp paired-end runs on a HiSeq 2500 (Illumina, San Diego, CA).

### Next-generation sequencing data analysis

For quality control and read mapping, paired-end reads were assessed for read quality using the FaQCs program [25]. Reads were trimmed when the per-base Phred score fell below 30 (Q30) and adaptor contamination was eliminated using FaQCs. The trimmed reads were aligned to the rhesus macaque genome sequence (Macaca mulatta GCF_000772875.2_Mmul_8.0.1) using HISAT2 (v2.0.6, default parameters) [26]. For differential expression analysis, reads aligned to annotated genes were counted and quantified using the htseq-count program [27]. The read counts were then used as input for pairwise identification of differentially expressed genes using DESeq2 [28]. Lists of differentially expressed genes were determined by filtering using an absolute log2 fold change cutoff of at least 2, *p*-value for fold change cutoff of at most 0.05, and the adjusted *p*-value (or false discovery rate, FDR) cutoff of at most 0.05. Volcano plots were prepared using a custom R script (ggplot2 library) [29]. Functional annotation including Gene Ontology (GO) terms and KEGG metabolic pathway analyses of the transcriptome data were performed with DAVID Bioinformatics Resources (v6.8) [30]. Overrepresented GO terms (in ontologies molecular function, biological process and cellular component) and KEGG pathways were identified with enrichment scores >1.3 with a *p*-value cutoff of 0.05.

## Results

### Transcriptome analysis in the liver of primary HEV gt1- and gt3-infected rhesus macaques by RNA sequencing

Transcriptome profiling was performed using the liver tissues from two native (baseline) and HEV gt1- and gt3-infected animals described previously [21]. The 10 animals, 3 with HEV gt1 and 7 with HEV gt3 infections, exhibited the typical course of acute viral hepatitis (Fig 1).

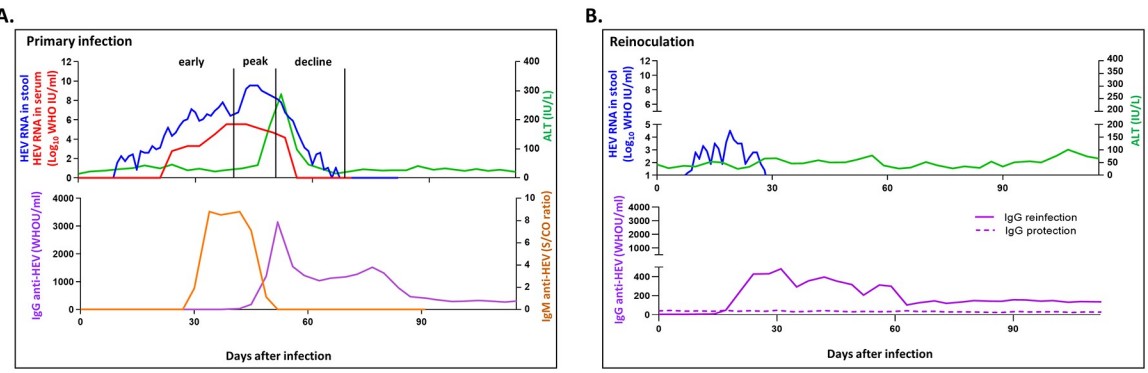

**Fig 1. Schematic diagram of course of HEV gt1 and gt3 infection in rhesus macaques.** The course of primary HEV gt1 and gt3 infection (A) and HEV reinfection with gt1 (B) have been described previously [21, 22]. Primary HEV genotype (gt) 1/gt3 infection and HEV reinfection were observed for 115 days after infection. Levels of HEV RNA in daily stool (blue line) and weekly serum (red line), and alanine aminotransferase (ALT) activities (green line) were shown as line graphs. Levels of IgM (brown line) and IgG (purple line) anti-HEV antibody responses were shown during primary HEV gt1/gt3 infection.

After HEV gt1 or gt3 inoculation, HEV RNA was detected from 2 to 9 days in stool and from 3 to 24 days in serum after inoculation. All animals had serum alanine aminotransferase (ALT) activity elevation from 35 to 42 days after inoculation and cleared the infection between 27 to 68 days after inoculation. Anti-HEV antibody responses became detectable from 12 to 43 days after inoculation [21, 22]. Viremia stages of primary HEV gt1- and gt3-infections were defined as early, peak, and decline phases based on HEV RNA titers in daily stool specimens (Fig 1 and Table 1).

For HEV gt1 infection, at the time of liver biopsy RH623 was in the early phase of infection, RH620 was at the peak of infection, and RH625 was in the decline phase of infection. In the HEV gt3 infection, two animals, RH654 and RH641 were in the early phase, 4 animals, RH644, RH639, RH645, and RH642 were at the peak of infection, and RH650 was in the decline phase. Approximately 46 million reads (±23 standard deviation) per sample were generated by RNA sequencing, and on average, 52% of the genes had non-zero counts (Fig 2A). Principal component analysis (PCA) showed that the liver sample used for the early, peak, and decline phases of HEV gt1 and gt3 infections were separated from each other (Fig 2B).

## Functional classification of differently expressed genes in HEV gt1 and gt3 infection

The transcriptomes of early, peak, and decline phases of primary HEV gt1 and gt3 infections were compared to two naïve control animals. A total of 415, 417, and 1769 differently expressed genes (DEGs) and 310, 678, and 388 DEGs were detected in early, peak and decline phases in the HEV gt1 and gt3 infections, respectively (fold change $\geq$ 2.0, $p \leq$ 0.05) (Fig 3A). In the HEV gt1 infection, 25% (n = 102, early), 30% (n = 123, peak), and 28% (n = 497, decline) of the DEGs were upregulated, whereas 74% (n = 310, early), 78% (n = 529, peak), and 46% (n = 177, decline) of the DEGs were upregulated in the HEV gt3 infection (Fig 3B). The highest percentages of DEGs were seen in different phases between the HEV gt1 and gt3 infections; 82% of total DEGs (n = 2193) detected in the decline phase of the gt1 infection and 66% of total DEGs (n = 678) regulated in the peak of gt3 infection (Fig 3B). 29 and 50 DEGs were co-expressed in all three phases of the gt1 and gt3 infections, respectively (Fig 3B).

In the HEV gt1 infection, significantly enriched GO analysis of DEGs were involved in the cellular response to zinc ion, isoprenoid, and glutathione biosynthesis in the early phase of

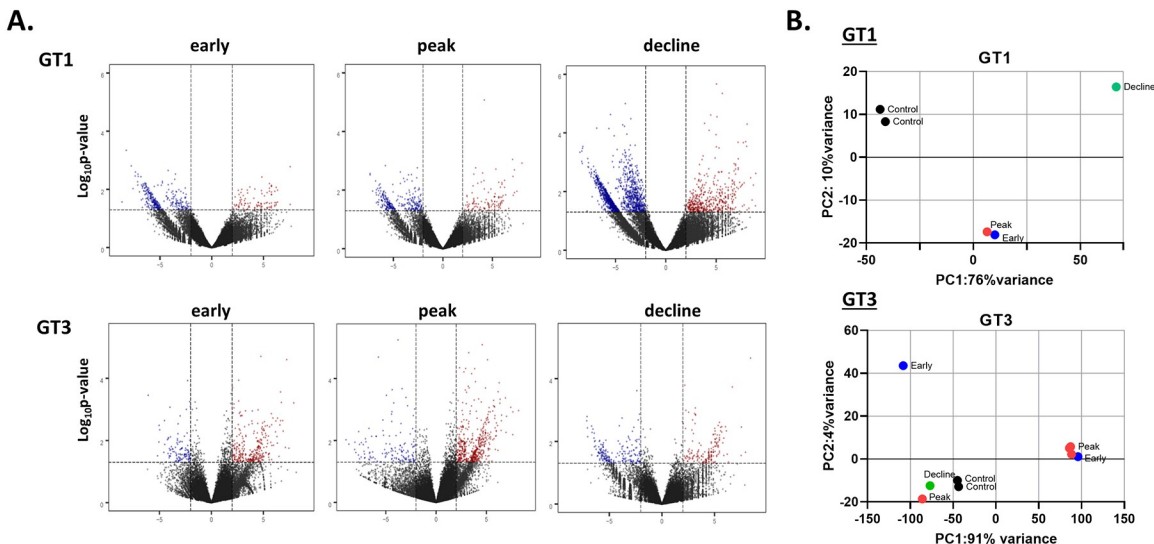

**Fig 2. Identification of differentially expressed genes (DEGs) in the liver tissues of experimentally infected rhesus macaques with HEV gt1 or gt3 infections.** (A), Volcano plot showing DEGs at early, peak, and decline phases of HEV gt1 and gt3 infections compared to two naïve control animals. Red-dot, upregulated genes; blue-dot, downregulated genes. (B), Principal component analysis (PCA) of each liver sample of HEV gt1 or gt3 infected animals and controls based on 33,372 unique DEGs identified at early, peak, decline and control samples.

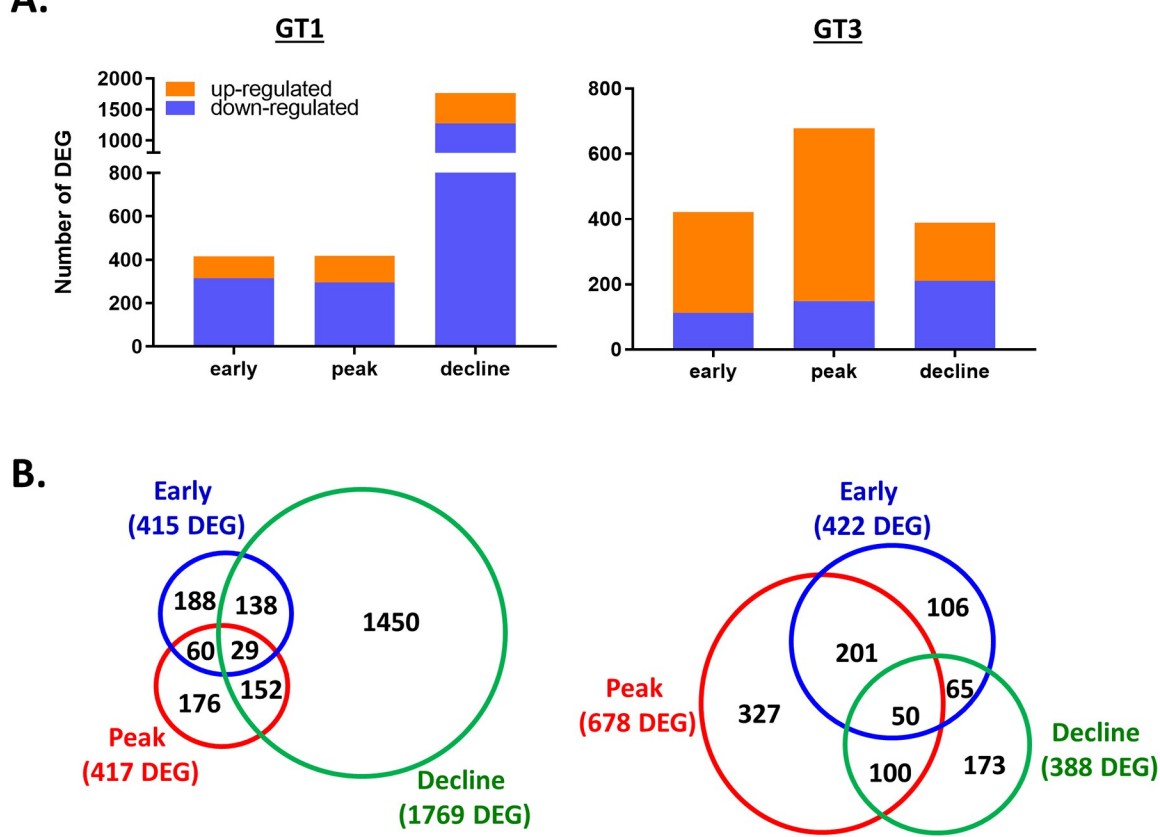

**Fig 3. Analysis of differentially expressed genes during HEV gt1 and gt3 infections.** (A), Bar chart of DEGs found to be upregulated or downregulated at early, peak, and decline phases during HEV gt1 and gt3 infections. (B), Venn diagram representing the number of DEGs between early, peak, and decline phases of HEV gt1 and gt3 infections.

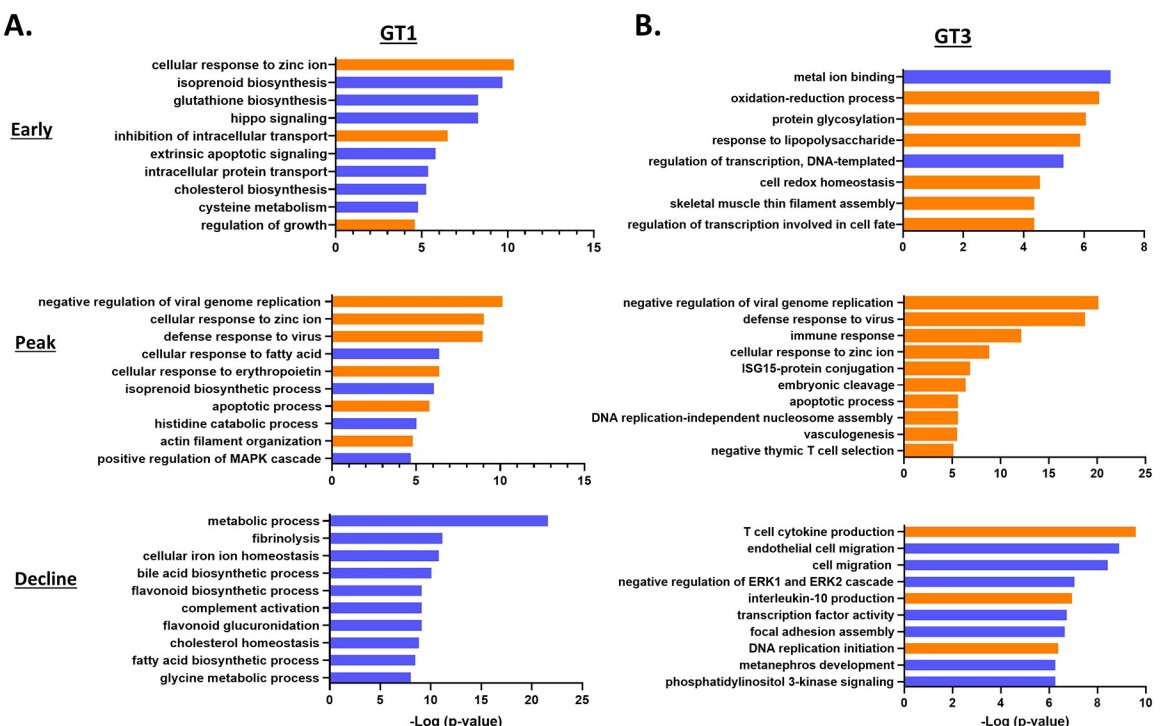

**Fig 4. Longitudinal differential GO term analysis during HEV gt1/gt3 infection.** Top 10 functional classification categories in HEV gt1 infection (A), gt3 infection (B). Overrepresented GO terms (in ontologies molecular function, biological process and cellular component) was identified with enrichment scores >1.3 with a p-value cutoff of > 0.05.

infection, negative regulation of viral genome replication, cellular response to zinc ion and defense response to virus at the peak of infection, and metabolic process, fibrinolysis, and cellular iron ion homeostasis in the gt1 infection decline phase (Fig 4A). In contrast, in the HEV gt3 infection, metal ion binding, oxidation-reduction process, and protein glycosylation DEGs were involved in the early phase, negative regulation of viral genome replication, defense response to virus and immune response at the peak, and T cell cytokine production, endothelial cell migration, and negative regulation of the ERK1 and ERK2 cascade process were significantly enriched in the decline phase of infection (Fig 4B).

## Pathway analysis of differently expressed genes against HEV gt1 and gt3 infection

Six, seven and eight of the top 10 predicted pathways at early, peak, and decline phases of the HEV gt1 infection were directly related to metabolic processes (Table 2).

Biosynthesis of antibiotics and metabolic pathways were significantly downregulated in all three phases of the gt1 infection. Highly regulated upstream genes were also found to be involved in the metabolic pathway (CYP4V2, HOGA1, GCLM, LYPLA1, BHMT, HMGCS1, ADA, MAOB, DHFR, and SULF2) (Fig 5A). In the HEV gt3 infection, pathways of ribosome, Huntington's disease, and oxidative phosphorylation in the early phase, pathways involved in various viral infections (measles, influenza, herpes simplex, and hepatitis C) at the peak, and transcriptional misregulation, and mineral absorption pathways in the decline phase were found to be significantly upregulated (Table 2). In addition, Rap1 signaling, cytochrome P450 metabolism, TGF-β and P13K-Akt signaling pathways were found to be down-regulated in the decline phase of the gt3 infection. Cell growth and differentiation (HRASLS2), transport

**Table 2. Predictive KEGG pathway of the differently expressed genes in each phase of primary HEV gt1 and gt3 infections and homologous HEV gt1 reinfection.**

| Term | Gene Count | P-Value | Term | Gene Count | P-Value |
|---|---|---|---|---|---|
| *GT1 early phase* | | | *GT3 early phase* | | |
| Biosynthesis of antibiotics | 14 | 0.00056 | Ribosome | 13 | 0.0016 |
| Lysosome | 8 | 0.0075 | Huntington's disease | 11 | 0.0022 |
| Terpenoid backbone biosynthesis | 4 | 0.0081 | Alzheimer's disease | 10 | 0.0045 |
| Sphingolipid signaling pathway | 8 | 0.0086 | Oxidative phosphorylation | 8 | 0.016 |
| Metabolic pathways | 35 | 0.025 | Non-alcoholic fatty liver disease | 8 | 0.018 |
| Glycolysis / Gluconeogenesis | 6 | 0.027 | Parkinson's disease | 8 | 0.022 |
| AMPK signaling pathway | 7 | 0.029 | Mineral absorption | 4 | 0.032 |
| Fatty acid degradation | 4 | 0.049 | | | |
| Chemical carcinogenesis | 5 | 0.05 | | | |
| *GT1 peak phase* | | | *GT3 peak phase* | | |
| Peroxisome | 9 | 0.00012 | Measles | 14 | 0.000018 |
| Metabolic pathways | 40 | 0.00032 | Influenza A | 14 | 0.00041 |
| Histidine metabolism | 5 | 0.0008 | Herpes simplex infection | 14 | 0.0015 |
| Fatty acid metabolism | 6 | 0.0012 | Mineral absorption | 6 | 0.0061 |
| Tryptophan metabolism | 6 | 0.0012 | Transcriptional misregulation in cancer | 11 | 0.0088 |
| Fatty acid degradation | 5 | 0.007 | Ribosome | 16 | 0.0091 |
| Biosynthesis of antibiotics | 11 | 0.0095 | Cell cycle | 9 | 0.016 |
| Protein processing in endoplasmic reticulum | 9 | 0.011 | Hepatitis C | 9 | 0.017 |
| Fc gamma R-mediated phagocytosis | 4 | 0.012 | | | |
| beta-Alanine metabolism | 4 | 0.023 | | | |
| *GT1 decline phase* | | | *GT3 decline phase* | | |
| Metabolic pathways | 203 | 3.6E-21 | Rap1 signaling pathway | 8 | 0.0036 |
| Complement and coagulation cascades | 31 | 6.7E-15 | Drug metabolism—cytochrome P450 | 4 | 0.018 |
| Biosynthesis of antibiotics | 54 | 3.4E-11 | Metabolism of xenobiotics by cytochrome P450 | 4 | 0.02 |
| Retinol metabolism | 22 | 2.8E-09 | Transcriptional misregulation in cancer | 5 | 0.028 |
| Chemical carcinogenesis | 24 | 0.000000016 | DNA replication | 3 | 0.029 |
| Valine, leucine and isoleucine degradation | 20 | 0.000000017 | Platelet activation | 5 | 0.033 |
| Ascorbate and aldarate metabolism | 12 | 0.000000024 | TGF-beta signaling pathway | 4 | 0.041 |
| Fatty acid degradation | 18 | 0.000000029 | PI3K-Akt signaling pathway | 8 | 0.041 |
| Drug metabolism—cytochrome P450 | 20 | 0.00000052 | Focal adhesion | 6 | 0.045 |
| Metabolism of xenobiotics by cytochrome P450 | 20 | 0.00000093 | NOD-like receptor signaling pathway | 3 | 0.05 |
| *reinfection* | | | *Protection* | | |
| Ribosome | 13 | 1.40E-03 | Ribosome | 25 | 1.90E-06 |
| Mineral absorption | 5 | 4.60E-03 | Mineral absorption | 8 | 2.40E-04 |
| Cell cycle | 3 | 0.058 | Huntington's disease | 13 | 2.40E-02 |
| FoxO signaling pathway | 3 | 0.058 | Alzheimer's disease | 12 | 3.30E-02 |
| | | | MAPK signaling pathway | 13 | 3.50E-02 |

Functional classification and pathways are ranked by the negative log of the p-value of the enrichment score.

(PNN, SLC4A4, and HBA1), metabolism (SRD5A3), and immune response (SSC4D) involved genes were highly regulated in the early HEV gt3 infection (Fig 5B). At the peak, the top upstream regulators were predominantly involved in the immune responses (IFI6, ISG15, MX1, IRF9, OAS2, and STAT1). Response to stress (MT1X), repair process (DCLRE1A), and T cell activation-related genes (B2M) were the most important upstream regulators during the decline phase of the gt3 infection. HRASLS2 was found to be upregulated in all three phases

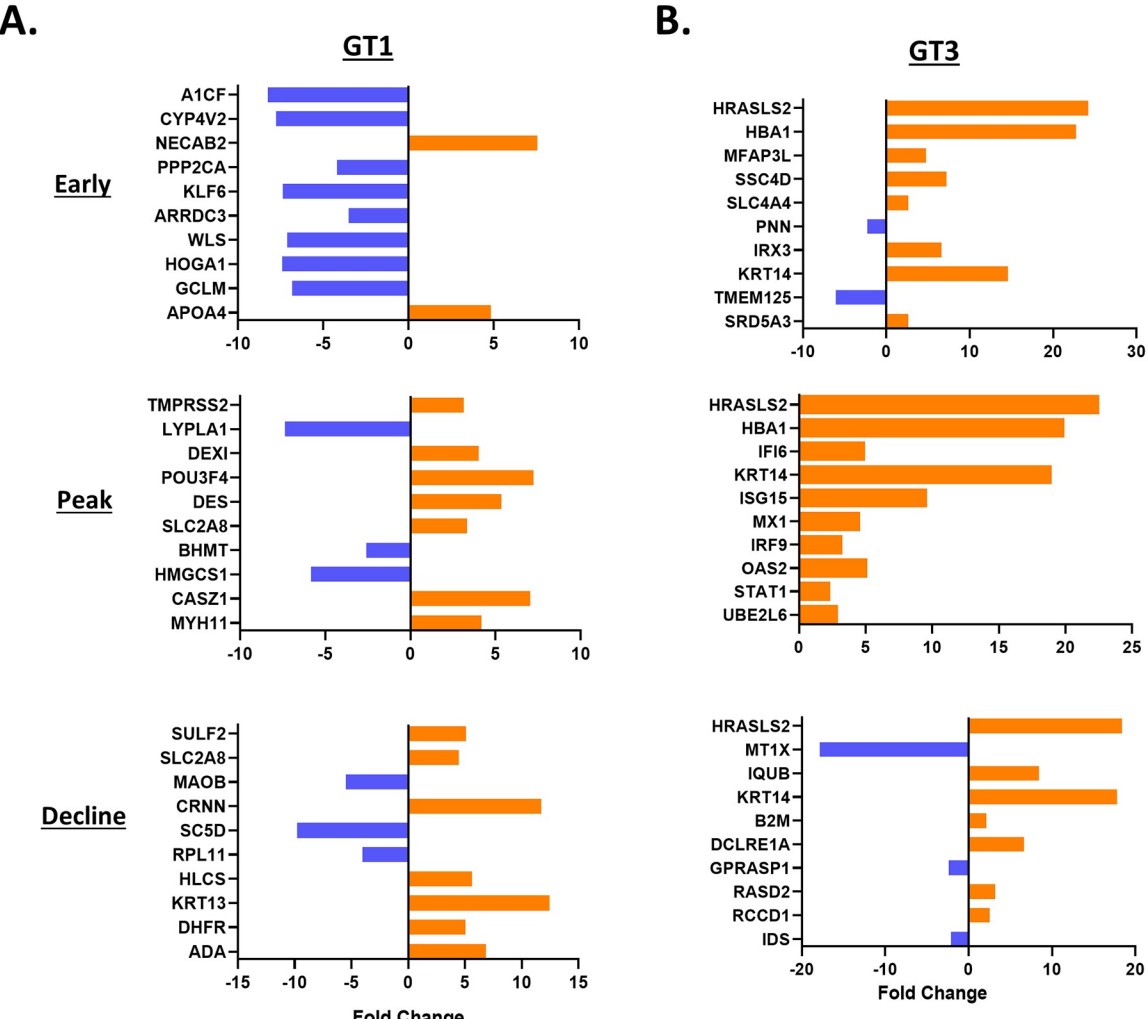

**Fig 5. Longitudinal differential gene expression analysis during HEV gt1/gt3 infection.** Top upstream 10 pathways predicted to be involved in HEV gt1 (A) and gt3 (B) infection. Functional classification and pathways are ranked by the negative log of the p-value of the enrichment score. Th significantly regulated genes ranked by the p-value and fold changes and identified with enrichment scores >1.3 with a p-value cutoff of > 0.05.

and upregulation of HBA1 gene was detected at the early and the peak phases of the HEV gt3 infection (Fig 5B).

## Transcriptome analysis after homologous HEV gt1 inoculation in seroconverted rhesus macaques

HEV RNA was observed in stools of 4 reinfected animals, average 17 days after re-inoculation. Liver biopsy samples from the re-infected animals were obtained during viral shedding (Table 2). The liver biopsy specimens from 5 protected animals with no observable viral shedding were taken at a similar time to that of the reinfected animals (Table 3).

RNA sequencing generated an average of 31 million reads (±37 standard deviation) in the re-infected and 30 million reads (±36 standard deviation) in the protected animals, respectively (Fig 6). A total of 370 significant DEGs were identified in the re-infected and 702 DEGs in the protected animals (fold change ≥ 2.0, p-value ≤ 0.05). 76% (n = 282) and 59% (n = 415) of the DEGs were upregulated in the reinfected and the protected animals, respectively

**Table 3. Liver biopsy sample of the animals with homologous HEV gt1 reinfection used for RNA sequencing.**

| HEV genotype | animal ID | Outcome after HEV gt1 re-inoculation | HEV RNA in liver (Log$_{10}$ WHO IU/ml) | Liver biopsy date | Time between 1st and 2nd infection (yr) |
|---|---|---|---|---|---|
| GT1 | RH637 | re-infection | 2.9 | 28 | 5.2 |
| GT1 | RH617 | re-infection | 4.3 | 19 | 2.8 |
| GT1 | RH636 | re-infection | 4.6 | 28 | 5.2 |
| GT1 | RH624 | re-infection | 2.1 | 19 | 2.8 |
| GT1 | RH621 | protection | NEG | 28 | 2.6 |
| GT1 | RH631 | protection | NEG | 19 | 2.4 |
| GT1 | RH625 | protection | NEG | 23 | 1.3 |
| GT1 | RH620 | protection | NEG | 23 | 1.2 |
| GT1 | RH627 | protection | NEG | 23 | 2.3 |

*, detection limit of HEV RNA by real-time PCR is 20 World Health Organization (WHO) IU/ml. GT1, genotype 1; NEG, negative detection of HEV RNA. The course of infection has been described previously (Choi et al., 2019). A pooled stool suspension from rhesus macaques with HEV gt1 infection, ranging from 2.24 to 3.24 Log$_{10}$ WHO IU/ml was used for re-inoculation (Choi et al, 201).

(Fig 6B). Principal component analysis (PCA) showed that the liver samples used for the reinfected were separated from those of protected animals (Fig 6C).

## Functional classification and pathway analysis of differently expressed genes in re-infection and protection against homologous HEV gt1 re-inoculation

In the HEV gt1 re-infection, significantly enriched GO Terms were involved in channel activity, cell cycle, immune responses, and cellular response to erythropoietin (Fig 7A). Highly

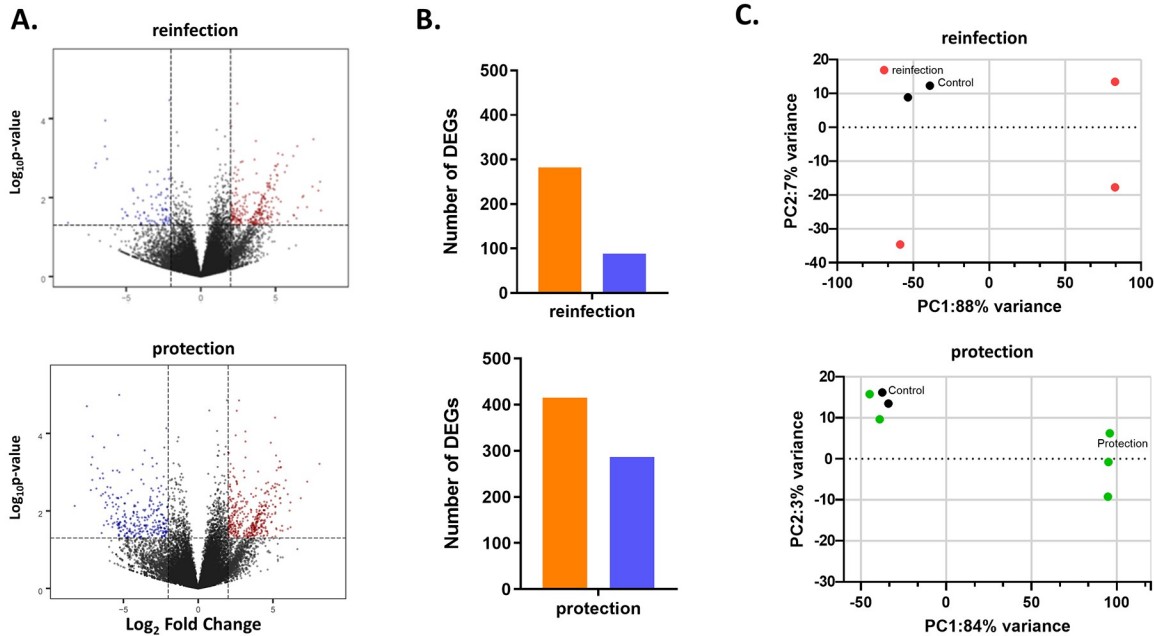

**Fig 6. Identification of differentially expressed genes (DEGs) in the liver tissues of seroconverted rhesus macaques with homologous HEV gt1 reinoculation.** (A), Volcano plot showing DEGs in reinfected and protected animals after HEV gt1 reinoculation compared to two naïve control animals. Red-dot, upregulated genes; blue-dot, downregulated genes. (B), Bar chart of DEGs found to be upregulated or downregulated in reinfected and protected animals after HEV gt1 inoculation. (C), Principal component analysis (PCA) of each liver sample of reinfected and protected animals after HEV gt1 inoculation and controls on the basis of 33,372 unique DEGs.

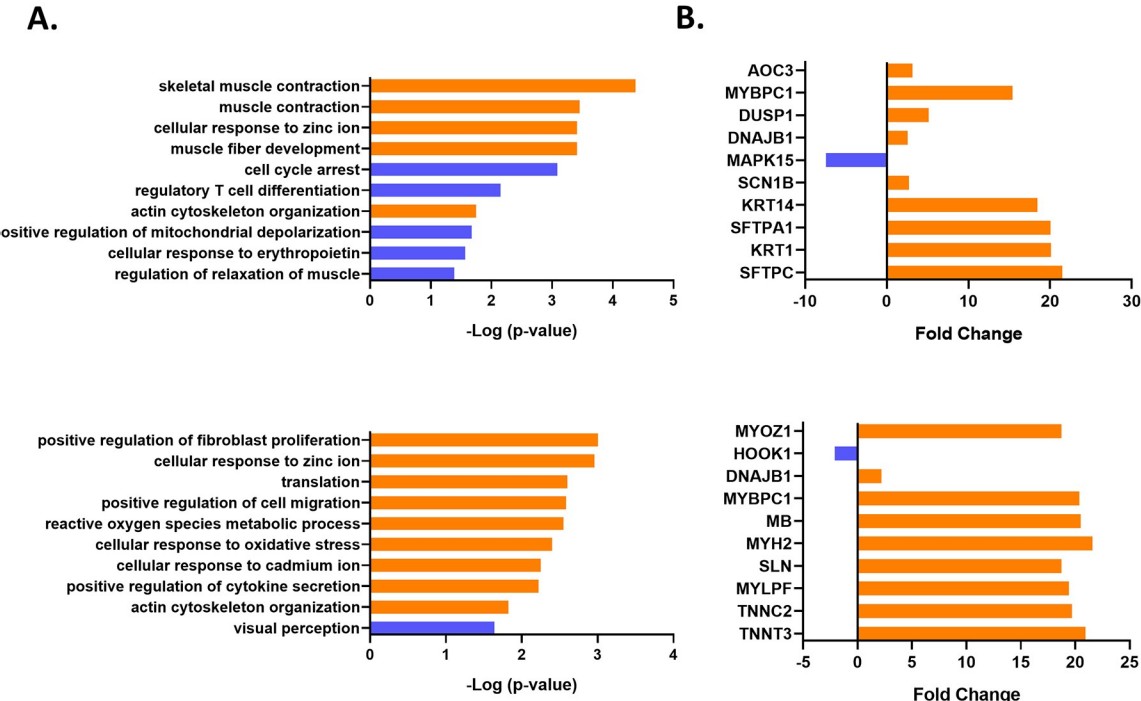

**Fig 7. Differential gene expression and pathway analysis after homologous HEV gt1 inoculation.** (A). Top 10 functional classification categories, and (B), Top 10 significantly regulated genes ranked by the p-value and fold change. Overrepresented GO terms (in ontologies molecular function, biological process and cellular component) were identified with enrichment scores >1.3 with a p-value cutoff of > 0.05.

enriched GO Terms in protection against HEV gt1 reinfection were fibroblast proliferation, channel activity, cellular response to oxidative stress, cytokine secretion, and cell migration. Pathway analysis of HEV gt1 re-inoculation revealed downregulation of cell cycle and FoxO signaling pathways in reinfected animals and upregulation of MAPK signaling pathway in protected animals (Table 2). Ribosome and mineral absorption pathways were predicted to be activated in both reinfection and protection against HEV re-inoculation. In HEV gt1 reinfection, significantly regulated genes were related to the immune responses (KRT1, DUSP1, and AOC3), signal transduction (MAPK15 and DNAJB1), transport (SFTP4), and metabolism (SFTPC) (Fig 7B). Top upstream regulators during protection against HEV gt1 after re-inoculation were predominantly involved in muscle contraction (MYH2, TNNT3, MYBPC1, TNNC2, and MYLPF), but several other top regulator genes were also found to be involved in transport (SLN and HOOK1) and signal transduction (MYOZ1 and DNAJB1).

## Discussion

In our previous study, the immune response-related gene expression profiles of HEV gt1 and gt3 infections were found to be significantly different in experimentally infected rhesus macaques [21]. To further define the longitudinal transcriptome profile of early, peak and decline phases of HEV gt1 and gt3 infections RNAseq analysis was employed with the goal of investigating genotype-specific molecular mechanisms of pathogenesis of HEV infection. This study provided the first hepatic transcriptome analysis of HEV gt1 and gt3 infections using experimentally infected rhesus macaques. Overall, HEV gt1 infection induced upregulation of 25%–30% DEGs, whereas 46%–78% of DEGs were upregulated in the HEV gt3 infection

(Fig 2A). Among total DEGs, 82% were expressed in the decline phase of gt1 infection and 66% were regulated at the peak phase of gt3 infection (Fig 2B). Similar observations were detected in our previous study where 25% of the hepatic immune response-related genes were downregulated in the gt1 infection, but 76% of genes were upregulated in the gt3 infection during the early phase of HEV RNA replication [21]. Analysis of DEGs indicates that more genes are upregulated than downregulated in HEV gt1 reinfection and protection against reinfection (Fig 5). This study further confirmed that the transcriptomes of HEV gt1 and gt3 infections were significantly different.

HEV gt1 and gt3 infections cause self-limited diseases but differ in geographical distributions and contrasting epidemiological characteristics [10]. In this study, we found that various biosynthesis processes (isoprenoid, glutathione, and cholesterol) and the metabolic pathways (glycolysis, fatty acid, tryptophan, and ascorbate) were significantly downregulated during the HEV gt1 infection, whereas host defense responses, apoptosis, and T cell cytokine production process were upregulated in the HEV gt3 infection (Figs 4 and 5; Table 2). These results suggest different host related anti-viral mechanisms were activated during HEV gt1 and gt3 infections where the metabolic pathways may have an important role in controlling HEV gt1 infection, and the host inflammatory immune responses in the HEV gt3 infection. Crosstalk between host immune responses and the metabolic system by viral infection have been shown in several different viral infections. For example, the innate immune response to mouse cytomegalovirus infection regulates the sterol metabolic pathway by downregulating sterol biosynthesis through an interferon regulatory loop [31]. The IFN signaling pathway activated by murine gammaherpesvirus-68 infection decreased cholesterol and fatty acid biosynthesis as a host defense mechanism [32]. In addition, hepatitis C virus (HCV) core protein can alter lipid metabolism through inhibition of microsomal triglyceride transfer protein and very low-density lipoprotein secretion [33]. These and our studies suggest that the activation of innate immune responses may allow downregulation of metabolic processes to serve as anti-viral immunity responses against HEV gt1 infection. Furthermore, HEV gt1 infection can lead to fulminant liver failure in pregnant women [13]. Pregnant women with fulminant hepatitis E were found to have the higher levels of steroid hormones and mortality rate [34]. Another study found that lower expression of progesterone receptors and high viral loads are responsible for the severity of HEV infection during pregnancy [35]. It is possible that the higher hormone levels could further influence metabolic pathways and immune suppression in pregnant women with HEV gt1 infection leading to high mortality rates.

Mitochondria are involved in several cellular metabolic processes and antiviral signaling pathways including the production of ATP and multiple neurodegenerative disorder pathogenesis such as Alzheimer's disease, Parkinson's disease, and Huntington's disease [36, 37]. In early HEV gt3 infection, genes associated with mitochondrial oxidative phosphorylation, Huntington's disease, Alzheimer's disease, and Parkinson's disease pathways were upregulated (Table 2), where genes in respiratory chain complex I (NDUFA1, NDUFA2, and NDUFS7), complex II (SDHC and SHHD), complex III (UQCRFS1), and complex V (ATP5D and COX5b) of oxidative phosphorylation were upregulated (S1 Table). The mitochondrial oxidative phosphorylation pathway is influenced by the duration and speed of the viral replication cycle [38]. Slow-replicating viruses like vaccinia, and rubella viruses' infection upregulated activity of respiratory chain complex I/ IV and complex II of mitochondrial oxidative phosphorylation, respectively and used mitochondria to maintain cellular energy homeostasis for their efficient replication (35, 36). We and others found the duration of HEV gt3 shedding in stool (26–50 days after infection) was shorter with lower titers (5.8 ± 0.3 log HEV RNA IU/g) than in the gt1 infection (55–67 days after infection; 6.8 ± 0.2 log HEV RNA IU/g) in rhesus macaques and humanized mice [21, 39]. These observations suggest that HEV gt3 may use

mitochondria to ensure energy for its replication during the early phase of the infection. In addition, extrahepatic manifestations of HEV infection have been identified where HEV can grow on a range of neurological cell lines, and HEV can cross the blood-brain barrier in experimentally infected animals [40]. HEV-associated acute encephalitic parkinsonism was also reported [41]. These results collectively suggest HEV gt3 infection influences mitochondrial dynamics to benefit early infection process and further affect extrahepatic viral pathogenesis.

In our previous study, type I IFN-response-related gene expression was upregulated in protected animals and levels of hepatic Th1/Th2 response-related gene expression were elevated in reinfected animals against homologous HEV gt1 reinfection [22]. In this study, mitogen-activated protein kinase (MAPK) signaling pathway was upregulated in the protected animals, and downregulation of cell cycle and the forkhead box O (FoxO) signaling pathways were detected in the reinfected animals (Table 2). The MAPK signaling pathways are initiated at the cell surface with the binding of specific ligands to various receptors. HCV core protein, the envelope protein, and HEV ORF3 as well as type I IFN response were reported to upregulate the MAPK signaling pathways [42–45]. In addition, oxidative stress activates MAPK signaling pathways [46]. Cellular response to oxidative stress was upregulated in the protected animals (Fig 7A). Taken together, these findings suggest that antiviral oxidative stress by HEV reinfection could induce upregulation of the MAPK signaling pathway to provide protection against secondary infection. The family of FoxO transcription factors plays an important role in cellular survival, death, proliferation and metabolism [47]. One of the FoxO transcription factors, FoxO3, was found to regulate CD8 T cell memory by T cell-intrinsic mechanisms [48]. During an acute lymphocytic choriomeningitis virus (LCMV) infection, FoxO3 controlled the accumulation of CD8 T cells by promoting cellular apoptosis, but FoxO3 deficiency dampened apoptosis of LCMV-specific CD8 T cells as well as increased numbers of memory CD8 T cells without compromising protective immunity against LCMV infection [48]. We speculate that downregulation of the FoxO signaling pathway might promote the high levels of HEV-specific CD8+ T cell responses to control HEV reinfection. The current study provides the first-time information on the differentially expressed hepatic transcriptome in HEV gt1/gt3 infections and gt1 reinfection in experimentally infected rhesus macaques, supporting hypothesis that HEV gt1 and gt3 infections induce transcriptome modifications contributing to disease pathogenesis. In addition, limited number of animals were used in primary HEV gt1/3 infections and gt1 reinfection, however, rhesus macaques have shown clinical signs consistent with acute viral hepatitis, viral shedding in feces and serum, and development of antiviral antibody responses that are similar to the natural history of HEV infection in humans and proved to be a reliable experimental model for studying HEV gt1/3 infections and gt1 reinfection [21, 22, 49, 50].

In conclusion, our study showed HEV gt1 infection induced regulation of metabolic pathways, and host defense responses were activated by HEV gt3 infection in rhesus macaques. This study provides further evidence that significantly different hepatic transcriptomes are induced to control pathogenesis during HEV gt1 and gt3 infections in experimentally infected rhesus macaques. In addition, HEV-specific T cell responses were activated to control in reinfection, while antiviral oxidative stress was induced in protection against HEV re-exposure.

## Supporting information

**S1 Table. Differently expressed genes in early phase of HEV gt1 infection.**
(XLSX)

**S2 Table. Differently expressed genes in peak phase of HEV gt1 infection.**
(XLSX)

**S3 Table. Differently expressed genes in decline phase of HEV gt1 infection.**
(XLSX)

**S4 Table. Differently expressed genes in early phase of HEV gt3 infection.**
(XLSX)

**S5 Table. Differently expressed genes in peak phase of HEV gt3 infection.**
(XLSX)

**S6 Table. Differently expressed genes in decline phase of HEV gt3 infection.**
(XLSX)

## Acknowledgments

**Disclaimers**: We thank animal caretakers (Comparative Medicine Branch, CDC) for caring the animals. This information is distributed under applicable information quality guidelines. It has not been formally disseminated by the Centers for Disease Control and Prevention/Agency for Toxic Substances and Disease Registry. It does not represent and should not be construed to represent any agency determination or policy. Use of trade names is for identification only and does not imply endorsement by the U.S. Department of Health and Human Services, the Public Health Service, or the Centers for Disease Control and Prevention. The findings and conclusions in this report are those of the authors and do not necessarily represent the views of the Centers for Disease Control and Prevention. No authors report conflicts of interest/ financial disclosures.

## Author Contributions

**Conceptualization:** Youkyung H. Choi.

**Data curation:** Youkyung H. Choi, Ganesh Srinivasamoorthy.

**Investigation:** Youkyung H. Choi, Xiugen Zhang.

**Supervision:** Michael A. Purdy.

**Visualization:** Youkyung H. Choi.

**Writing – original draft:** Youkyung H. Choi.

**Writing – review & editing:** Youkyung H. Choi, Michael A. Purdy.

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
