## [Decision Letter · Decision Letter 0]

30 Jun 2020

PONE-D-20-13527

Transcriptome analysis in rhesus macaques infected with hepatitis E virus genotype 1/3 infections and genotype 1 re-infection

PLOS ONE

Dear Dr. Choi,

Thank you for submitting your manuscript to PLOS ONE. After careful consideration, we feel that it has merit but does not fully meet PLOS ONE’s publication criteria as it currently stands. Therefore, we invite you to submit a revised version of the manuscript that addresses the points raised during the review process.

specifically, as requested by the two reviewers please explain more in details why only 3 animals exposed to HEV-1 were included when there were 7 in the animals exposed to HEV-3. As noted by one of them some methods used were not described in the material and method, please add the corresponding sub-chapter with corresponding publication references.

We look forward to receiving your revised manuscript.

Kind regards,

Pierre Roques, Ph.D.

Academic Editor

PLOS ONE

Journal Requirements:

Reviewers' comments:

Reviewer's Responses to Questions

**Comments to the Author**

1. Is the manuscript technically sound, and do the data support the conclusions?

Reviewer #1: Yes

Reviewer #2: Yes

2. Has the statistical analysis been performed appropriately and rigorously? 

Reviewer #1: Yes

Reviewer #2: Yes

3. Have the authors made all data underlying the findings in their manuscript fully available?

Reviewer #1: Yes

Reviewer #2: Yes

4. Is the manuscript presented in an intelligible fashion and written in standard English?

Reviewer #1: Yes

Reviewer #2: Yes

5. Review Comments to the Author

Reviewer #1: Minor revisions

Material and Methods

Line 82 Please, inform if the research protocol was approved by Institutional Revision Board for laboratory animals.

Results

Line 97 to100 “A liver biopsy was performed aseptically once on two animals prior to

hepatitis E virus inoculation to observe baseline liver tissue for host response induced by hepatitis E virus infection using a one-port incisional abdominal entry method with a laparoscope”.

Question: The pre inoculation liver samples were collected to transcriptome analysis, so these transcripts data should be described individually, comparing pre and post HEV infection (HEV1 or HEV3 infections).

Line 158 “Transcriptome profiling was performed using the liver tissues from HEV gt1- and gt3-infected animals described previously (21). The 10 animals, 3 with HEV gt1 and 7 with HEV gt3 infections, exhibited the typical course of acute viral hepatitis (Fig 1).

Question: The figure 1 could be more informative. The authors may show laboratory parameters for HEV 1 re-infected group in the same graph type of HEV infected monkeys, divided in protected and not protected subgroups data.

Line 199 to 203 “The transcriptomes of early, peak, and decline phases of primary HEV gt1 and gt3 infections were compared to two naïve control animals. A total of 415, 417, and 1769 differently expressed genes (DEGs) and 310, 678, and 388, 200 DEGs were detected in early, peak and decline phases in the HEV gt1 and gt3 infections, respectively (fold change201 ≥ 2.0, p ≤ 0.05) (Fig 3A).

Question: Why the re-infected group (protected and not protected) were not included in analysis expressed in Figure 3 and 4?

Discussion

Line 341-343: “We and others found the duration of HEV gt3 shedding in stool (26-50 days after infection)was shorter with lower titres (5.8 ± 0.3 log HEV RNA IU/g) than in the gt1 infection (55-67 days after infection; 6.8343 ± 0.2 log HEV RNA IU/g) in rhesus macaques and humanized mice (21, 39)”.

Question: I`m not sure about this find, because data analysis did not confirm significance of this conclusion.

Finally, based in differences (host response) presented here the authors may open discussion if HEV genotype 1 and 3 should be classified as being the same hepatitis E virus.

Reviewer #2: Manuscript Number: PONE-D-20-13527

Article Type: Research Article

Full Title: Transcriptome analysis in rhesus macaques infected with hepatitis E virus genotype

1/3 infections and genotype 1 re-infection

Short Title: Transcriptome analysis in HEV gt1/gt3 infection

Authors: Youkyung H. Choi, Ph.D. Centers for Disease Control and Prevention Atlanta, GA UNITED STATES.

The study presents the results of primary scientific research and the results reported here have not been published elsewhere. The manuscript describes a technically sound piece of scientific research with data that supports the conclusions, with appropriate replication. The conclusions are appropriately based on the data presented.

The design of the experiments and materials and methods are described in sufficient detail with high technical standard.

The article is presented in an intelligible fashion and is written in standard English.

The research meets all applicable standards for the ethics of experimentation and research integrity. The authors explained all animal work conducted according to relevant national and international guidelines: Federal regulatory requirements for animal care and use and were approved by the Institutional Animal Care and Use Committee at the Centers for Disease Control and Prevention (CDC) in Atlanta, GA. The authors also described procedures in order to minimize the stress of the animals throughout the study period and the animals were assessed for clinical signs associated with complications associated with the liver biopsy procedure.

The statistical analysis has been performed appropriately and rigorously.

The manuscript follows the PLOS Data policy. All data are fully available without restriction.

HEV causes both epidemic and sporadic viral hepatitis. HEV is classified into eight genotypes but only 1 to 4 infects humans. These 4 genotypes have different geographic distribution and contrast epidemiologically, but also clinical picture and evolution. Therefore molecular mechanisms of pathogenesis related to genotype-specific HEV infection is important to understand the disease, therapeutic options, HEV-associated extrahepatic manifestations, among others.

This study provided the first hepatic transcriptome analysis of HEV gt1 and gt3 infections using experimentally infected rhesus macaques. These results suggest different host related anti-viral mechanisms were activated during HEV gt1 and gt3 infections where the metabolic pathways may have an important role in controlling HEV gt1 infection, and the host inflammatory immune responses in the HEV gt3 infection.

Using new technology the study showed that HEV gt1 infection induced regulation of metabolic pathways, and host defense responses were activated by HEV gt3 infection in rhesus macaques. In addition, provides further evidence that significantly different hepatic transcriptomes are induced to control pathogenesis during HEV gt1 and gt3 infections in experimentally infected rhesus macaques. Finally the study shows that HEV-specific T cell responses were activated to control in reinfection, while antiviral oxidative stress was induced in protection against HEV re-exposure.

The design and the laboratory aspects relative to laparoscopic and needle liver biopsy liver biopsy procedures, inoculum and blood sampling, sample preparation and next-generation sequencing data analysis were describes in detail.

Comments on detailed sections.

Materials and Methods

Page 6, Inoculum and blood sampling

Three rhesus macaques (RH623, RH620, and RH625) were inoculated intravenously with a pooled stool suspension from rhesus macaques infected with the human HEV gt1 Sar-55 strain. Seven animals (RH654, RH641, RH644, RH639, RH645, RH642, and RH650) were inoculated intravenously with a pooled stool suspension from rhesus macaques infected with a human HEV gt3 strain.

- Why do you use 3 for gt 1 and 7 for gt 3?

In the results the authors used real time PCR, antibodies detection and ALT activity, but the methodology is not described in Materials and Methods.

Page 9, Results

Figure 2. Identification of differentially expressed genes (DEGs) in the liver tissues of experimentally infected rhesus macaques with HEV gt1 or gt3 infections. (A), Volcano plot showing DEGs at early, peak, and decline phases of HEV gt1 and gt3 infections compared to two naïve control animals.

Naïve control animals; please described the use of controls animals properly in Materials and Methods.

6. PLOS authors have the option to publish the peer review history of their article (what does this mean?). If published, this will include your full peer review and any attached files.

Reviewer #1: No

Reviewer #2: No

---

## [Author Response · Author response to Decision Letter 0]

23 Jul 2020

Editorial Comments:

specifically, as requested by the two reviewers please explain more in details why only 3 animals exposed to HEV-1 were included when there were 7 in the animals exposed to HEV-3. As noted by one of them some methods used were not described in the material and method, please add the corresponding sub-chapter with corresponding publication references.

We inoculated six macaques (RH623, RH633, RH620, RH625, RH635, and RH634) with HEV gt1. However, we found that RH633, RH635, and RH634 were co-infected with simian immunodeficiency virus (SIV). For this reason, these three animals were excluded from the study.

Missing methods with corresponding publication references were added as sub-chapters in Materials and Methods.

Response to the Reviewer 1:

Reviewer #1: Minor revisions

Material and Methods

Line 82 Please, inform if the research protocol was approved by Institutional Revision Board for laboratory animals. 

Thank you for the reviewer’s comment. Institutional Revision Board at CDC reviews only the studies involved in humans. Studies using laboratory animals are reviewed by the Institutional Animal Care and Use Committee (IACUC) at the Centers for Disease Control and Prevention (CDC). This study was approved by CDC IACUC. This is described in lines 68 to 69.

Results

Line 97 to100 “A liver biopsy was performed aseptically once on two animals prior to

hepatitis E virus inoculation to observe baseline liver tissue for host response induced by hepatitis E virus infection using a one-port incisional abdominal entry method with a laparoscope”.

Question: The pre inoculation liver samples were collected to transcriptome analysis, so these transcripts data should be described individually, comparing pre and post HEV infection (HEV1 or HEV3 infections).

According to the reviewer’s suggestion, descriptions of the liver samples of native control and HEV-infected animals were added in Materials and Methods (line 146). In addition, description of native control was added in Results (lines 25 and 170).

Line 158 “Transcriptome profiling was performed using the liver tissues from HEV gt1- and gt3-infected animals described previously (21). The 10 animals, 3 with HEV gt1 and 7 with HEV gt3 infections, exhibited the typical course of acute viral hepatitis (Fig 1).

Question: The figure 1 could be more informative. The authors may show laboratory parameters for HEV 1 re-infected group in the same graph type of HEV infected monkeys, divided in protected and not protected subgroups data.

According to the reviewer’s suggestion, figure 1 is divided into primary infection and reinoculation of HEV. HEV gt1 reinoculation of HEV is differentiated into reinfection (not protected) and protection.

Line 199 to 203 “The transcriptomes of early, peak, and decline phases of primary HEV gt1 and gt3 infections were compared to two naïve control animals. A total of 415, 417, and 1769 differently expressed genes (DEGs) and 310, 678, and 388, 200 DEGs were detected in early, peak and decline phases in the HEV gt1 and gt3 infections, respectively (fold change201 ≥ 2.0, p ≤ 0.05) (Fig 3A).

Question: Why the re-infected group (protected and not protected) were not included in analysis expressed in Figure 3 and 4?

Thank you for reviewer’s comment. Reinoculation of HEV was performed with only gt1 virus. We felt that a comparison between primary HEV g1 and gt3 infection would be more appropriate to analyze differently regulated host transcriptomes. In addition, analyzing transcriptome in primary HEVgt1 and gt3 infection would provide better understanding on the relationship between genotype-specific epidemiologic characteristics and molecular mechanisms of pathogenesis.

Discussion

Line 341-343: “We and others found the duration of HEV gt3 shedding in stool (26-50 days after infection)was shorter with lower titres (5.8 ± 0.3 log HEV RNA IU/g) than in the gt1 infection (55-67 days after infection; 6.8343 ± 0.2 log HEV RNA IU/g) in rhesus macaques and humanized mice (21, 39)”.

Question: I`m not sure about this find, because data analysis did not confirm significance of this conclusion.

We thank the reviewer for pointing this out. Differences between HEV gt1 and gt3 infection in rhesus macaques are unknown. Based on our study, we did not detect the differences in clinical disease markers such as ALT activity, viral shedding, and antibody responses between gt1 and gt3 infection in experimentally infected rhesus macaques. However, differently regulated host genes in early HEV gt3 infection were found to be associated with mitochondrial oxidative phosphorylation, Huntington’s disease, Alzheimer’s disease, and Parkinson’s disease pathways, whereas metabolic pathways were highly regulated in HEV gt1 infection. We speculated that our observation on difference in duration of the viral shedding and levels of HEV gt3 viral titer may contribute to these differences in host transcriptomes in HEV gt3 infection.

Finally, based in differences (host response) presented here the authors may open discussion if HEV genotype 1 and 3 should be classified as being the same hepatitis E virus.

We thank the reviewer for pointing this out. Even if HEV gt1 and gt3 infections have genotype-specific epidemiological characteristics as well as the significantly different patterns of hepatic gene expression, both gt1 and gt3 cause liver disease in human and are classified as a member of the genus Orthohepevirus A within the Hepeviridae family. We felt that discussing classification of HEV gt1 and gt3 is beyond scope of our study. 

Response to the Reviewer 2:

Reviewer #2: Manuscript Number: PONE-D-20-13527

Article Type: Research Article

Full Title: Transcriptome analysis in rhesus macaques infected with hepatitis E virus genotype

1/3 infections and genotype 1 re-infection

Short Title: Transcriptome analysis in HEV gt1/gt3 infection

Authors: Youkyung H. Choi, Ph.D. Centers for Disease Control and Prevention Atlanta, GA UNITED STATES.

The study presents the results of primary scientific research and the results reported here have not been published elsewhere. The manuscript describes a technically sound piece of scientific research with data that supports the conclusions, with appropriate replication. The conclusions are appropriately based on the data presented.

The design of the experiments and materials and methods are described in sufficient detail with high technical standard.

The article is presented in an intelligible fashion and is written in standard English.

The research meets all applicable standards for the ethics of experimentation and research integrity. The authors explained all animal work conducted according to relevant national and international guidelines: Federal regulatory requirements for animal care and use and were approved by the Institutional Animal Care and Use Committee at the Centers for Disease Control and Prevention (CDC) in Atlanta, GA. The authors also described procedures in order to minimize the stress of the animals throughout the study period and the animals were assessed for clinical signs associated with complications associated with the liver biopsy procedure.

The statistical analysis has been performed appropriately and rigorously.

The manuscript follows the PLOS Data policy. All data are fully available without restriction.

HEV causes both epidemic and sporadic viral hepatitis. HEV is classified into eight genotypes but only 1 to 4 infects humans. These 4 genotypes have different geographic distribution and contrast epidemiologically, but also clinical picture and evolution. Therefore molecular mechanisms of pathogenesis related to genotype-specific HEV infection is important to understand the disease, therapeutic options, HEV-associated extrahepatic manifestations, among others.

This study provided the first hepatic transcriptome analysis of HEV gt1 and gt3 infections using experimentally infected rhesus macaques. These results suggest different host related anti-viral mechanisms were activated during HEV gt1 and gt3 infections where the metabolic pathways may have an important role in controlling HEV gt1 infection, and the host inflammatory immune responses in the HEV gt3 infection.

Using new technology the study showed that HEV gt1 infection induced regulation of metabolic pathways, and host defense responses were activated by HEV gt3 infection in rhesus macaques. In addition, provides further evidence that significantly different hepatic transcriptomes are induced to control pathogenesis during HEV gt1 and gt3 infections in experimentally infected rhesus macaques. Finally the study shows that HEV-specific T cell responses were activated to control in reinfection, while antiviral oxidative stress was induced in protection against HEV re-exposure.

The design and the laboratory aspects relative to laparoscopic and needle liver biopsy liver biopsy procedures, inoculum and blood sampling, sample preparation and next-generation sequencing data analysis were describes in detail.

Comments on detailed sections.

Materials and Methods

Page 6, Inoculum and blood sampling

Three rhesus macaques (RH623, RH620, and RH625) were inoculated intravenously with a pooled stool suspension from rhesus macaques infected with the human HEV gt1 Sar-55 strain. Seven animals (RH654, RH641, RH644, RH639, RH645, RH642, and RH650) were inoculated intravenously with a pooled stool suspension from rhesus macaques infected with a human HEV gt3 strain.

- Why do you use 3 for gt 1 and 7 for gt 3?

We inoculated six macaques (RH623, RH633, RH620, RH625, RH635, and RH634) with HEV gt1. However, we found that RH633, RH635, and RH634 were co-infected with simian immunodeficiency virus (SIV). For this reason, these three animals were excluded from the study.

In the results the authors used real time PCR, antibodies detection and ALT activity, but the methodology is not described in Materials and Methods.

According to the reviewer’s suggestion, sections of “determination of HEV RNA titer by real-time PCR” and “detection of anti-HEV antibodies and ALT activity” were added into Materials and Methods.

Page 9, Results

Figure 2. Identification of differentially expressed genes (DEGs) in the liver tissues of experimentally infected rhesus macaques with HEV gt1 or gt3 infections. (A), Volcano plot showing DEGs at early, peak, and decline phases of HEV gt1 and gt3 infections compared to two naïve control animals.

Naïve control animals; please described the use of controls animals properly in Materials and Methods. According to the reviewer’s suggestion, descriptions of the liver samples of native control and HEV-infected animals were added in Materials and Methods (line 146). In addition, description of native control was added in Results (lines 25 and 170).

---

## [Decision Letter · Decision Letter 1]

30 Jul 2020

Transcriptome analysis in rhesus macaques infected with hepatitis E virus genotype 1/3 infections and genotype 1 re-infection

PONE-D-20-13527R1

Dear Dr. Choi,

We’re pleased to inform you that your manuscript has been judged scientifically suitable for publication and will be formally accepted for publication once it meets all outstanding technical requirements.

Kind regards,

Pierre Roques, Ph.D.

Academic Editor

PLOS ONE

Additional Editor Comments (optional):

Reviewers' comments:

Reviewer's Responses to Questions

**Comments to the Author**

1. If the authors have adequately addressed your comments raised in a previous round of review and you feel that this manuscript is now acceptable for publication, you may indicate that here to bypass the “Comments to the Author” section, enter your conflict of interest statement in the “Confidential to Editor” section, and submit your "Accept" recommendation.

Reviewer #1: All comments have been addressed

2. Is the manuscript technically sound, and do the data support the conclusions?

Reviewer #1: Yes

3. Has the statistical analysis been performed appropriately and rigorously? 

Reviewer #1: Yes

4. Have the authors made all data underlying the findings in their manuscript fully available?

Reviewer #1: Yes

5. Is the manuscript presented in an intelligible fashion and written in standard English?

Reviewer #1: No

6. Review Comments to the Author

Reviewer #1: In my opinion the author's responses were enough. I don't have any additional comments to be done about the current manuscript.

7. PLOS authors have the option to publish the peer review history of their article (what does this mean?). If published, this will include your full peer review and any attached files.

Reviewer #1: No

---

## [Editor Report · Acceptance letter]

24 Aug 2020

PONE-D-20-13527R1 

 Transcriptome analysis in rhesus macaques infected with hepatitis E virus genotype 1/3 infections and genotype 1 re-infection 

Dear Dr. Choi:

I'm pleased to inform you that your manuscript has been deemed suitable for publication in PLOS ONE. Congratulations! Your manuscript is now with our production department. 

Kind regards, 

on behalf of

Dr. Pierre Roques 

Academic Editor

PLOS ONE